# Glioblastoma Stem-like Cell Detection Using Perfusion and Diffusion MRI

**DOI:** 10.3390/cancers14112803

**Published:** 2022-06-04

**Authors:** Tanguy Duval, Jean-Albert Lotterie, Anthony Lemarie, Caroline Delmas, Fatima Tensaouti, Elizabeth Cohen-Jonathan Moyal, Vincent Lubrano

**Affiliations:** 1ToNIC, Toulouse NeuroImaging Center, Université de Toulouse, Inserm, UPS, 31000 Toulouse, France; lotterie.ja@chu-toulouse.fr (J.-A.L.); fatima.tensaouti@inserm.fr (F.T.); vincent.lubrano@inserm.fr (V.L.); 2Department of Nuclear Medicine, CHU Purpan, 31000 Toulouse, France; 3U1037 Toulouse Cancer Research Center CRCT, INSERM, 31000 Toulouse, France; lemarie.anthony@iuct-oncopole.fr (A.L.); moyal.elizabeth@iuct-oncopole.fr (E.C.-J.M.); 4Université Paul Sabatier Toulouse III, 31000 Toulouse, France; 5Institut Claudius Regaud, IUCT-Oncopole, 31000 Toulouse, France; delmas.caroline@iuct-oncopole.fr; 6Service de Neurochirurgie, Clinique de l’Union, 31240 Toulouse, France

**Keywords:** glioblastoma, MRI, surgery, stem cells, FLAIR, perfusion, diffusion

## Abstract

**Simple Summary:**

Glioblastoma stem-like cells (GSCs) are known to be aggressive and radio-resistant and proliferate heterogeneously in preferred environments. Additionally, quantitative diffusion and perfusion MRI biomarkers provide insight into the tissue micro-environment. This study assessed the sensitivity of these imaging biomarkers to GSCs in the hyperintensities-FLAIR region, where relapses may occur. A total of 16 patients underwent an MRI session and biopsies were extracted to study the GSCs. In vivo and in vitro biomarkers were compared and both Apparent Diffusion Coefficient (ADC) and relative Cerebral Blood Volume (rCBV) MRI metrics were found to be good predictors of GSCs presence and aggressiveness.

**Abstract:**

Purpose: With current gold standard treatment, which associates maximum safe surgery and chemo-radiation, the large majority of glioblastoma patients relapse within a year in the peritumoral non contrast-enhanced region (NCE). A subpopulation of glioblastoma stem-like cells (GSC) are known to be particularly radio-resistant and aggressive, and are thus suspected to be the cause of these relapses. Previous studies have shown that their distribution is heterogeneous in the NCE compartment, but no study exists on the sensitivity of medical imaging for localizing these cells. In this work, we propose to study the magnetic resonance (MR) signature of these infiltrative cells. Methods: In the context of a clinical trial on 16 glioblastoma patients, relative Cerebral Blood Volume (rCBV) and Apparent Diffusion Coefficient (ADC) were measured in a preoperative diffusion and perfusion MRI examination. During surgery, two biopsies were extracted using image-guidance in the hyperintensities-FLAIR region. GSC subpopulation was quantified within the biopsies and then cultivated in selective conditions to determine their density and aggressiveness. Results: Low ADC was found to be a good predictor of the time to GSC neurospheres formation in vitro. In addition, GSCs were found in higher concentrations in areas with high rCBV. Conclusions: This study confirms that GSCs have a critical role for glioblastoma aggressiveness and supports the idea that peritumoral sites with low ADC or high rCBV should be preferably removed when possible during surgery and targeted by radiotherapy.

## 1. Introduction

Glioblastoma is the most aggressive primary brain tumor whose current standard treatment associates maximal safe surgical resection, followed by radiotherapy (RT) and Temozolomide (TMZ) based chemotherapy combination [1]. Nonetheless, despite aggressive treatment, glioblastoma is not curable. All patients present a recurrence that mostly occurs close to the surgical cavity and within the volume targeted by radiation therapy [2,3]. Median survival is 15 months [4].

Such an outcome is due both to glioblastoma propensity for infiltration, degrading surgical resection efficiency, and extensive tumoral heterogeneity resulting in variable biological responses to radio-chemotherapy. Despite being a minor population of cancer cells, the cancer stem cells (CSCs) that are identified in glioblastoma (GSCs) are thought to be the major driving force behind glioblastoma biological heterogeneity and are likely to explain the high rates of glioblastoma recurrence. Indeed, GSCs are particularly aggressive and radio-resistant [5]. Moreover, they have the ability to self-renew rapidly in vitro to form neurospheres [6], and show great plasticity which could explain the high heterogeneity of glioblastoma [7] and GBM recurrence after radiotherapy [8,9]. All glioblastomas contain CSC, but unfortunately the preferential localization of these cells in the tumor volume in vivo is unknown due to the absence of studies on the sensitivity of medical images to highlight this specific cell population.

On MRI scan, glioblastoma clearly appears as a heterogeneous tissue. The typical structural imaging features of a glioblastoma include an infiltrative, heterogeneous, ring-enhancing lesion with central necrosis and surrounding peritumoral edema. In clinical routines, two contrasts are commonly used to image glioblastomas: a contrast-enhanced (CE) T1-weighted image that reveals active regions which indicate blood–brain barrier breakdown and a FLAIR (Fluid Attenuated Inversion Recovery) image that highlights other tissue abnormalities in the peritumoral region (e.g., tissue softening, water infiltration, and vasogenic edema). If we hypothesized that GSCs preferentially develop in specific microenvironment conditions, then a particular MRI contrast should be able to predict their presence in the peritumoral edema component. MR spectroscopic imaging (MRSI) is a method sensitive to specific molecules that provide information about glioblastoma metabolism. In particular, choline (Cho), a marker of membrane synthesis, increases in glioblastoma due to tumorous cell proliferation [10], and *N*-acetyl-aspartate (NAA), a neuronal marker, decreases due to neuronal cell loss and dysfunction [11,12]. MRSI is classically used to show the infiltrative nature of glioblastoma in the non-CE [13], and the total volume of elevated choline (Cho) to N-acetylaspartate (NAA) index (CNI = Cho/NAA) has been associated with less favorable survival [14]. In the same way, a ratio of CNI above two was also found to be a predictor of the site of relapse in patients with glioblastoma [15]. However, the low signal to noise ratio of MR spectroscopy and its sensitivity to magnetic field inhomogeneity make MR spectroscopy imaging particularly challenging and time-consuming in clinical practice (resolution is in the order of 1 × 1 × 1 cm^3^) [16]. Reduced apparent diffusion coefficient (ADC), measured with diffusion MRI [17], is observed in patients with glioblastoma due to an increased cellularity [18]. ADC is lower in higher grade glioma, which can be used to discriminate glioma subtypes [19]. In general, ADC is correlated with reduced overall survival of the patients [20,21] and is a predictor of early recurrence [22]. Elevated cerebral blood volume (CBV), measured with perfusion MRI, is also observed in patients with high-grade glioma [23]. Notably, CBV has been shown to differentiate recurrent tumors from pseudo-progression with high sensitivity and specificity (90%) [24]. Multimodal MRI, where diffusion and perfusion metrics are combined, improves the prediction of recurrence [25], progression-free survival, and overall survival [26]. From these results, we can argue that diffusion and perfusion MRI could affect a particular environment allowing the survival of more resistant GBM cells to radio-chemotherapy.

In this proof-of-principle study, we wondered whether peritumoral regions with reduced diffusion and elevated perfusion could be associated with a higher concentration of GSC. To this end, biopsy samples were extracted in the peritumoral edema region with MR-image guidance during resection surgery and analyzed by flow cytometry. Sites of biopsy were strategically targeted using the independent biomarker CNI in order to maximize the chance to achieve a biopsy in regions with high tumoral activity. Finally, the extracted cells were cultivated in vitro. The proportion of GSC in the biopsies was counted in a cytometer and the time for glioblastoma-extracted cells to grow as GSC-enriched neurospheres in vitro was reported, the latter being a surrogate marker of tumor cell aggressiveness as shown in patients [27] and in xenografted mice [28]. These two measurements were correlated with ADC and CBV maps in order to determine the sensitivity of these metrics to GSC concentration and aggressiveness.

## 2. Methods

Figure 1 illustrates the overall methodology of the study. An imaging session with anatomical, diffusion, perfusion, and spectroscopy MRI was performed one week before surgery. Then, during surgery, biopsy samples were extracted from the peritumoral edema component of the tumors using intraoperative MR image guidance.

### 2.1. Ethics Statement

Experiments in this study refer to the prospective clinical trial STEMRI NCT01872221. Study Protocol and consent form has received ethics approval from the “Comité de Protection des Personnes Sud-Ouest et Outre Mer III” (N° 2012-A00585-38) on 31 May 2012. The ethical principles for medical research involving human subjects of the WMA declaration of Helsinski were followed to obtain informed and written consent from the patients to participate in the STEMRI study.

### 2.2. Patient Population

Sixteen patients with newly diagnosed GBM were enrolled in a prospective pilot study trial for newly diagnosed glioblastoma amenable to surgical resection.

Inclusion criteria were (i) patients with a primary diagnosis of glioblastoma brain tumor, (ii) 18 years old or more, (iii) no psychiatric background, and (iv) written consent from the patient to participate in the STEMRI study. Patients were deemed suitable for undergoing complete resection of the contrast-enhancing tumor by the neurosurgeon (VL.) with the goal of achieving maximum tumor resection. Table 1 summarizes the patient population characteristics.

### 2.3. MR Imaging Data Acquisition

Up to a week before surgery, MRI images were acquired on a 3T MRI system (ACHIEVA dStream, Philips Healthcare, Best, The Netherland). A 32-channel phased-array receive coil was used. The following MRI sequences were acquired:

Anatomical. The anatomical MRI protocol included:T1CE. 3D T1-weighted after 15 mL injection of Gadolinium contrast (TR/TE = 8/4 ms, FA = 8°, matrix = 165 × 241, 240 slices, 1 × 1 × 1 mm^3^ resolution).FLAIR (TI/TR/TE = 2400/8000/335 ms, FA = 90°, matrix = 200 × 256, 256 slices, resolution = 1 × 1 × 1 mm^3^)T2w. Turbo-spin echo T2-weighted (TR/TE = 4130/80 ms, FA = 90°, matrix = 512 × 512, 43 slices, resolution = 0.5 × 0.5 × 3 mm^3^).

Diffusion MRI. Diffusion-weighted data were acquired with a single shot EPI sequence. Parameters were: TR = 10 s, b-value = 1000 s/mm^2^, 15 directions (+1 b = 0), TE = 55 ms, matrix = 112 × 112, voxel size = 2 × 2 × 2 mm^3^, 60 slices.

Perfusion: A dynamic susceptibility contrast-enhanced (DSC) MRI sequence was used with a spin-echo echo-planar imaging sequence. TR/TE = 1700/50 ms, FA = 75°, matrix size = 128 × 128, 22 slices, voxel size = 1.75 × 1.75 × 5 mm^3^.

Spectroscopy. 2D Spectroscopy was performed using a point-resolved spectroscopy acquisition (PRESS) over 4 slices covering the lesions. Matrix = 8 × 8, voxel size = 1 × 1 × 1 cm^3^, TR/TE = 1000/144 ms, two averages.

Total acquisition time was around 45–50 min. The field of view was shifted away from brain–air interfaces when necessary.

### 2.4. MR Imaging Data Processing

Motion correction, image registration to the anatomical T1w image, fiber tracking from diffusion images (DTI-FT), Apparent Diffusion Coefficient (ADC) and Cerebral Blood Flow (CBV) maps were computed using the Sisyphe Toolbox developed at our center [30]. Normalization of the CBV (rCBV) was performed using a manually selected white matter region in the controlateral part of the tumor (CBV_WM_): rCBV = CBV/CBV_WM_. The median values of ADC and rCBV were then extracted at the biopsy location.

Biopsy density is defined by the needle size of 1.8 mm. The precision of the location is defined by (i) the spatial registration of the neuronavigation system (4 mm) [31], (ii) the stability of the needle during extraction (approximately 2 mm), and (iii) the resolution of the anatomical image used for guidance (1 mm, leading to a precision of (1/√12)mm at 1σ). Finally, the biopsy lies in a volume of 680 mm^3^ at 1σ, or 2040 mm^3^ at 3σ, around the estimated location.

Starting from the voxel corresponding to the estimated biopsy location, a morphological dilation was applied as far as the region of interest (ROI) reached 2040 mm^3^ to take into account these uncertainties. This dilatation was constrained in the manually drawn hyperintensities-FLAIR region.

### 2.5. Surgical Planning and Definition of Targets

Image segmentation: The tumors were manually segmented using preoperative MRI scan, on the anatomical images (T1CE, FLAIR), in two components: tumor bulk (CE and necrosis) and tumor edema (FLAIR). Two additional segmentations were performed in the hyperintensities-FLAIR according to the CNI > 2 and CNI < 2 regions following the results of [15]. DTI-FT was used to extract the motor corticospinal tract (CST), the dorsal language stream (arcuate and superior longitudinal fasciculus (SLF)) and the ventral language stream (inferior fronto-occipital (IFO) and uncinatus (UNC)) [32,33].

The segmented tumor volumes, fiber tracking, and computed maps (ADC and rCBV) were transferred to the neuronavigation planning system BrainLab iPlan, and the operating room BrainLab VectorVision Neuronavigation System (Nnav).

Preparation: Prior to surgery, the patient and family were counseled by the surgeon, a physician assistant, and an anesthesiologist. Patients were operated under general anesthesia or awake surgery, depending on the location of the tumor and functional criticality. Patient positioning for the surgery took into consideration neck comfort, unobstructed vision for continuous neurologic and language assessment, unobstructed line of sight for infrared cameras used for stereotaxis, and access to the airway in case emergency intubation was required. Intraoperative brain-mapping techniques (i.e., cortico-subcortical electrostimulation) were used in addition to non-invasive tractography to spare essential sites subserving motor and language control. The head of the patient was fixed in a Mayfield head-clamp to maximize stereotactic accuracy.

Surgery and biopsy extraction: Prior to resection, in order to minimize brain shift, two biopsies were performed according to the CNI > 2 and <2 in the peritumoral edema regions using intraoperative MRI guidance. Resection was then performed according to the surgical standards of maximum safe resection of CE and more if possible, using neuronavigation and refined to take into account electro-stimulation. Early postoperative MRI scans were obtained as soon as possible after surgery and no later than 72 h after surgery.

### 2.6. Stem Cell Culture and Analysis

After extraction, biopsies were processed using our established protocol [8] and a small fraction of the sample was directly analyzed by flow cytometry to study the percentage of the GSC subpopulation (GSC%) over the total number of cells, as previously described [8]. Then, remaining cells were cultured as GSC-enriched neurospheres (NS) in stem cell medium, as described in Ref. [8]. In these restrictive conditions, only GSCs can survive and form NS. All cultivated samples were observed daily under a microscope to check for the appearance of NS and thus report the time to neurosphere formation in days.

The GSC% is a marker of malignant stem-like cells’ infiltration and the time to neurosphere formation is associated with stem-like cells’ aggressiveness. Figure 2 shows the two sites of biopsy in each patient on the T1w images.

### 2.7. Statistics

ADC and rCBV were correlated using Pearson method to GSC% and to the time to neurosphere formation.

## 3. Results

### 3.1. Multiparametric MRI Maps

Due to a failure of the gadolinium injector, the perfusion sequence could not be run in four different patients. rCBV could not be computed, and these data were not included in the correlations of rCBV with biopsy metrics.

Figure 3 shows the rCBV and ADC quantitative maps in each patient. We qualitatively observe (i) similar values in the normal white matter between patients, consistent values of ADC around 0.5–1 × 10^−3^ mm^2^/s [34], and rCBV values about twice as large in gray matter (rCBV~2) as in white matter (rCBV~1), and (ii) good quality data with no artefacts and good SNR.

### 3.2. Cell Culture

In two subjects, biopsies were of insufficient quality to be analyzed and were therefore discarded. In one of the other 14 subjects, one biopsy sample could not be retrieved, leading to a total of 27 biopsy samples analyzed (two samples in each subject). Note that the three of them were part of the group with injector failure (see previous paragraph). In addition, for 7 of the 27 biopsy samples, the neurospheres had not grown after three months. In total, we retrieved 27 measures of GSC% and 20 measures of the time to neurosphere formation from 14 patients.

### 3.3. Correlations between MRI and Biology

Figure 4 shows the correlation between imaging metrics and sample characterization. rCBV was associated with the proportion of GSC (GSC%) counted in the biopsies (24 data points, r^2^ = 0.17, *p* = 0.04) and ADC with time to neurospheres (TTN) formation in vitro (20 data points, r^2^ = 0.5, *p* = 0.0003). rCBV versus TTN and ADC versus GSC% correlations were low (r^2^ < 0.12) and not significant (*p* > 0.05). These results suggest that ADC is specifically related to GSC aggressiveness and that rCBV areas are enriched in GSC.

For illustration, a representative patient is shown in Figure 5. Biopsy location and associated ROIs are delineated. MRI (rCBV, ADC) and biopsies’ characterization values are presented.

## 4. Discussion

In this study, we showed a strong relationship between in vivo advanced MRI metrics and the in vitro biological behavior of CSC.

Sixteen patients were recruited, and data from 14 of them could be analyzed. This number is consistent with the recommendations (nine patients for detecting a 10% change in ADC) [35] due to the high sensitivity and precision of ADC and rCBV metrics to abnormalities.

Biopsies were extracted in the peritumoral edema in order to assess the MR sensitivity to infiltrative GSC. Although a similar study could be performed in the contrast-enhanced ring, such a region is less homogeneous, relatively thin (less than a centimeter) and close to the necrosis, making the extraction more sensitive to biopsy location and thus less precise. Note that the MR guidance system is calibrated on the skull, and a small shift of the brain with respect to the skull is inevitable. CNI was used as an independent index to prevent any selection bias of the biopsy location while maximizing the chance of high tumoral activity [15].

We show for the first time that ADC obtained from diffusion MRI was inversely related to tumor cells’ aggressiveness as measured by the time to neurosphere formation from human biopsies. These regions with lower edema and dense cell packing appear more suitable for the most aggressive GSCs. This result corroborates studies showing the relationship between ADC and Progression Free Survival (PFS) [36,37].

Although previous studies have shown higher diffusion in regions with higher GSC concentration identified through CD44 marker [38], we did not find this correlation (r^2^ < 0.001) in our study. This result corroborates the high variability of ADC values in Glioblastoma found by Ref. [39]. A competing effect seemed to occur with denser cellularity due to cell proliferation (thus decreasing the diffusion coefficient) and higher permeability of stem cell membranes due to high AQP4 expression and activity [40], especially in Glioblastoma [41], thus increasing the ADC values. Interestingly, chemoradiation has been shown to increase diffusion coefficient and fade ADC contrast with respect to the contralateral part of the brain [37].

rCBV was correlated with GSC percentage which could be explained by the fact that GSCs are preferentially localized near vascular niches [42,43,44,45] known to allow their self-renewal [46]. This result in the peritumoral edema (hyperintensities-FLAIR region) is consistent with the finding of higher rCBV in the CE region [39,47]. Correlation between rCBV and the time to neurosphere formation was not significant (r^2^ < 0.001), suggesting that rCBV is not a good predictor of GSC aggressiveness.

The intratumoral heterogeneity of GBM accounts for many persisting clinical challenges in diagnosis and treatment planning and the extension of the resection to non CE is still under debate. While resecting CE has been shown to remain the main goal, with no detectable effect on PFS with blind resection of the hyperintensities-FLAIR region [48], it has also been shown that extending resection using specific MRI [49] or fluorescence [50] biomarkers does have an impact. Indeed, peritumoral edema is heterogenous [51] and is infiltrated by cancer cells and extending resection in these areas may prolong survival [52]. Furthermore, major edema was found to be a poor prognostic factor [53].

However, the peritumoral edema often sustains critical functions (e.g., motricity, sensitivity, language) and cannot be resected blindly. As the FLAIR hyperintensities areas are heterogeneous and cannot be removed in totality, our results show that higher rCBV and lower ADC value in these areas should be also removed when possible or used to optimize the delineation of target volume in radiotherapy. Indeed, the current routine and standard recommendations for radiotherapy planning delineation only use conventional MRI sequences [54].

## 5. Conclusions

In this work, to our knowledge, we show for the first time that reduced ADC is a good predictor of tumor cell aggressiveness, while high rCBV reveals the localization of potential niches of glioblastoma stem cells. Our results point out that addition of advanced MRI improves glioblastoma heterogeneity knowledge and could improve patient-specific surgical and radiotherapy planning.

## Figures and Tables

**Figure 1 cancers-14-02803-f001:**
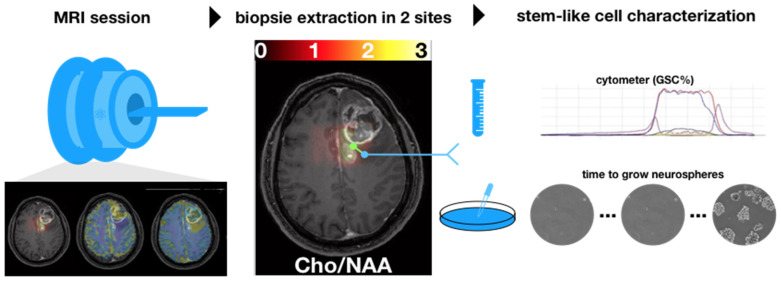
A multimodal MRI session was prescribed up to one week before the surgery. During the surgery, two biopsies were extracted in the hyperintensities-FLAIR region based on the ratio cho/NAA (CNI) (threshold set to 2). Each biopsy was characterized by using a cytometer and by cultivating stem-like cells.

**Figure 2 cancers-14-02803-f002:**
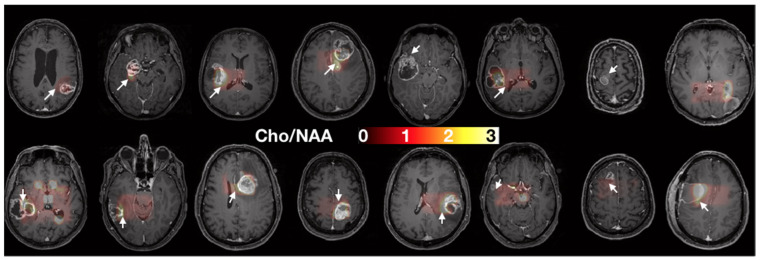
Axial slices of each patient prior to surgery. Spectroscopy (Cho/NAA) maps overlying T1 contrast-enhanced images were used to determine the two sites of sample collection (i.e., one target in the CNI > 2 region, white arrows, and one target control in the CNI < 2 regions. When no CNI > 2 was present, the maximal CNI value was used for the target.

**Figure 3 cancers-14-02803-f003:**
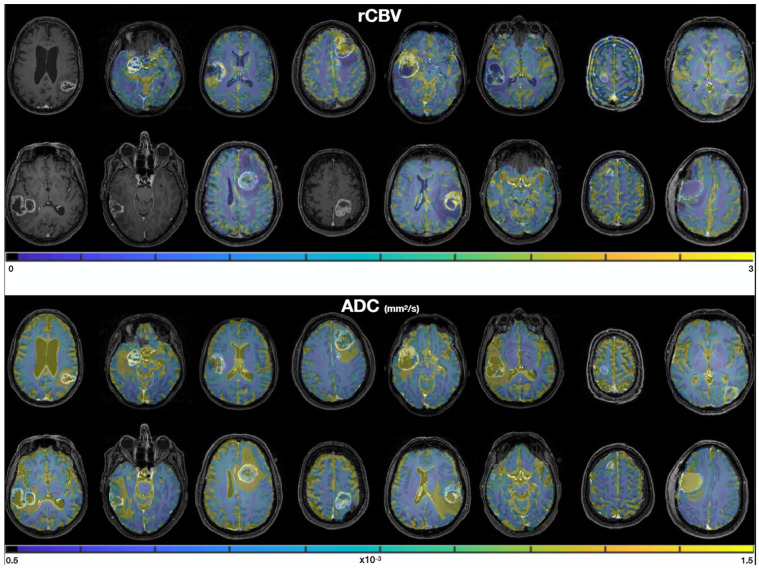
rCBV (**top**) and ADC (**bottom**) maps, overlayed on the T1 contrast-enhanced images. Masks of 2000 px were drawn in the two biopsy locations in order to extract median rCBV and ADC values for each sample.

**Figure 4 cancers-14-02803-f004:**
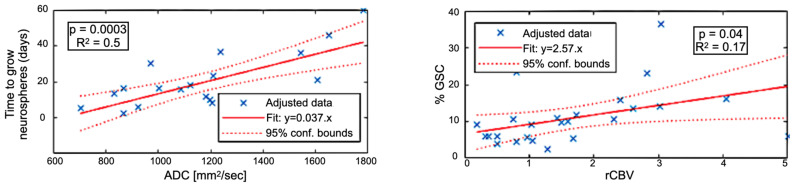
(**Left**) Correlation between the time to grow GSC neurospheres (in days) versus the ADC. (**Right**) Correlation between the proportion of GSC versus the relative Cerebral Blood Volume (rCBV).

**Figure 5 cancers-14-02803-f005:**
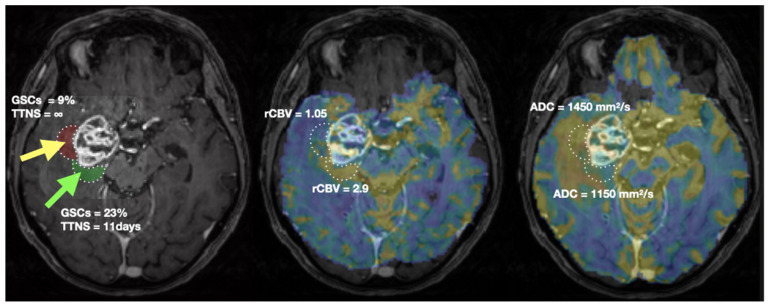
Biopsies location (green arrow: CNI > 2, yellow arrow: CNI < 2) and characterisation (**left**), rCBV (**middle**) and ADC (**right**) values in a representative patient. The two ROIs of 2040 mm^3^ used to extract MRI metrics are shown in green and red on the T1CE image (**left**). TTNS: Time To NeuroSphere formation.

**Table 1 cancers-14-02803-t001:** Population characteristics.

Patient	Sex	Age	FLAIR Volume (cm^3^)	Contrast-Enhanced Volume (cm^3^)
sub-001	M	76	60	23
sub-002	F	76	37	9
sub-003	M	66	92	8
sub-006	F	73	66	14
sub-007	M	59	151	36
sub-008	F	57	37	22
sub-009	F	72	39	19
sub-010	M	52	114	23
sub-012	M	62	166	15
sub-013	F	60	92	32
sub-014	M	72	26	1
sub-015	M	73	79	17
sub-017	M	49	22	2
sub-019	M	78	14	3
sub-020	M	37	90	10
sub-021	M	56	31	11
MEDIAN		64	63	15
MIN		376	14	1
MAX		787	166	36

Progression was evaluated according to RANO criteria [29].

## Data Availability

The data presented in this study is available in this article.

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
