# Peer review of "Glioblastoma Stem-like Cell Detection Using Perfusion and Diffusion MRI"

_cancers, 2022, doi:10.3390/cancers14112803_

Round 1
Reviewer 1 Report
To editors and reviewers
Glioblastoma stem-like cell detection using perfusion and diffusion MRI
- This is an very interesting manuscript that can be considered for publication in CANCERS. The manuscript is appropriate with aims and scope of journal.
- I suggested some revisions below and after revisions the manuscript can be published.
1) Some citation and references are not precise as MDPI format. Please check and revise.
2) All figures should include icons such as arrow or arrowhead to indicate lesion.
3) Figure 4 had bad resolution. Please try to enhance it.
4) The conclusion should have only one paragraph. Please revise.
5) Some essential and updated documents related to CNS neoplasms in discussion should be checked such as
PMID: 32588986
PMID: 32366451
Sincerely
Reviewer 2 Report
In this study, image-guidance targeted biopsies were extracted intraoperatively from 16 glioblastoma patients in the hyper-FLAIR region and GSC subpopulation was quantified within the biopsy and then cultivated in selective conditions to determine their density and aggressiveness. They found that low ADC was found to be a good predictor of the time to GSC neurospheres formation in vitro, and that GSCs were in higher concentrations in locations with high rCBV on perfusion. They concluded that GSCs have a critical role for glioblastoma aggressiveness and peritumoral sites with low ADC or high rCBV should be preferably removed when possible during surgery and targeted by radiotherapy.
About the detection of GSCs,
What antibodies were used to detect GSCs?
The authors should show the results of the percentage of GSCs in representative cases.
If they use CD133 or CD44 to detect GSCs, were there any difference in results between CD133-positive GSCs and CD44-positive GSCs?
The authors measured the percentage of GSGs in total cells obtained by biopsy. Higher CBV lesion is considered to contain more tumor cells, compared to lower CBV lesion. Higher percentage of GSCs in total cells might be due to the higher percentage of tumor cells in the total cells. Are there any difference in the percentage of GSCs in total tumor cells obtained biopsy between higher CBV lesion and lower CBV lesions?
If there was any data, it is better to add the information of the percentage of GSCs in the enhance lesions.
About the tumor growth,
Was the time to grow the mean time or the minimum time?
The authors described that the time to grow in vitro condition was associated with stem-like cell’s aggressiveness. If so, please add the reference to prove it.
Author Response
Please see the attachement.

Round 2
Reviewer 2 Report
The manuscript has been revied well. I think this manuscript will be acceptable.